# Molecular Characterization of the First Alternavirus Identified in *Fusarium oxysporum*

**DOI:** 10.3390/v13102026

**Published:** 2021-10-08

**Authors:** Caiyi Wen, Xinru Wan, Yuanyuan Zhang, Hongyan Du, Chenxing Wei, Rongrong Zhong, Han Zhang, Yan Shi, Jiatao Xie, Yanping Fu, Ying Zhao

**Affiliations:** 1College of Plant Protection, Henan Agricultural University, Zhengzhou 450002, China; wencaiyi@henau.edu.cn (C.W.); xinruwan@126.com (X.W.); eryazhang5@163.com (Y.Z.); 19971117@163.com (H.D.); chenxingWS@163.com (C.W.); zhongrongrong2021@126.com (R.Z.); hanzhang1996@126.com (H.Z.); shiyan00925@126.com (Y.S.); 2State Key Laboratory of Agricultural Microbiology, Huazhong Agricultural University, Wuhan 430070, China; jiataoxie@mail.hzau.edu.cn (J.X.); yanpingfu@mail.hzau.edu.cn (Y.F.)

**Keywords:** *fusarium oxysporum*, alternavirus, “*Alternaviridae”*, mycovirus, hypovirulence

## Abstract

A novel mycovirus named Fusarium oxysporum alternavirus 1(FoAV1) was identified as infecting *Fusarium oxysporum* strain BH19, which was isolated from a fusarium wilt diseased stem of *Lilium brownii*. The genome of FoAV1 contains four double-stranded RNA (dsRNA) segments (dsRNA1, dsRNA 2, dsRNA 3 and dsRNA 4, with lengths of 3.3, 2.6, 2.3 and 1.8 kbp, respectively). Additionally, dsRNA1 encodes RNA-dependent RNA polymerase (RdRp), and dsRNA2- dsRNA3- and dsRNA4-encoded hypothetical proteins (ORF2, ORF3 and ORF4), respectively. A homology BLAST search, along with multiple alignments based on RdRp, ORF2 and ORF3 sequences, identified FoAV1 as a novel member of the proposed family “*Alternaviridae**”*. Evolutionary relation analyses indicated that FoAV1 may be related to alternaviruses, thus dividing the family “*Alternaviridae”* members into four clades. In addition, we determined that dsRNA4 was dispensable for replication and may be a satellite-like RNA of FoAV1—and could perhaps play a role in the evolution of alternaviruses. Our results provided evidence for potential genera establishment within the proposed family “*Alternaviridae”*. Additionally, FoAV1 exhibited biological control of Fusarium wilt. Our results also laid the foundations for the further study of mycoviruses within the family “*Alternaviridae”*, and provide a potential agent for the biocontrol of diseases caused by *F. oxysporum*.

## 1. Introduction

Mycoviruses are a group of viruses that infect filamentous fungi, yeasts and oomycetes. Until recently, the National Center of Biotechnology Information (NCBI) database contained more than 300 mycoviral sequences, divided into at least 19 families by the International Committee for Taxonomy of Viruses (ICTV) [1]. Mycoviruses contain double-stranded RNA (dsRNA), positive single-stranded RNA (+ssRNA), negative single-stranded RNA (-ssRNA) and single-stranded DNA (ssDNA) as their genome [2,3,4]. Double-stranded RNA (dsRNA) mycoviruses are classified into seven families: *Totiviridae, Endornaviridae, Partitiviridae, Chrysoviridae, Megabirnaviridae, Quadriviridae* and *Reoviridae*, and one genus, *Botybirnavirus* [5,6].

Many mycoviruses affect fungal virulence and some can cause alterations in host phenotypes, reducing or enhancing virulence [2,7,8,9,10,11,12]. This property of mycoviruses, which reduces the ability of their fungal hosts to cause disease, is termed hypovirulence [2,4]. Hypovirulent mycoviruses have the potential to elicit biological control of plant pathogenic fungal diseases [2,4,12]. Typical examples of hypovirulent mycovirus application include Cryphonectria hypovirus 1 (CHV1), which is used to control chestnut blight caused by *Cryphonectria parasitica* in Europe, and the Sclerotinia sclerotiorum hypovirulence-associated DNA virus 1 (SsHADV-1), which is used to control stalk break caused by *Sclerotiorum sclerotiorum* [13,14,15,16]. In recent years, many hypovirulence-associated mycoviruses have been found in other plant pathogenic fungi, such as *Botryosphaeria cinerea* debilitation-related virus (BcDRV), Botryosphaeria dothidea Chrysovirus 1 (BdCV1), Rhizoctonia solani partitivirus 2 (RSPV2), Rhizoctonia solani endornavirus 1 (RsEV1) and Colletotrichum liriopes partitivirus 1 (ClPV1) [17,18,19,20].

Further studies have revealed many novel viruses that differ obviously from already-known viruses. For example, Hammond et al. (2008) reported a novel mycovirus, *Aspergillus* mycovirus 341 (AsV341), from *Aspergillus niger* [21]. The closest known relative of the AsV341 was a totivirus, Sphaeropsis sapinea RNA virus 2 (SsRV2), from *Sphaeropsis sapinea* [21]. SsRV2 contains only one genomic RNA segment (5202 bp) and is a typical totivirus, based on the sequence characteristics identified and phylogenetic analysis with other known totiviruses [22]. Moreover, AsV341 contains four segments (between 1.5 and 3.6 kbp), and the amino acid identity and similarity levels between AsV341 and SsRV2 are only 15.2% and 27.8%, respectively [21]. Thus, AsV341 is obviously different from the known totivirus, even though its closest relative is SsRV2 [21]. Aoki et al. (2009) reported a novel mycovirus, Alternaria alternata virus 1 (AaV1), from *Alternaria alternata*, which is similar to AsV341, following an analysis of its sequence characteristics and multiple alignment. However, phylogenetic analysis showed that AaV1 and AsV341 did not cluster with any mycovirus family [23]. In 2013, Kozlakidis et al. [24] reported a novel mycovirus (Aspergillus foetidus virus (AfV)), which was similar to both AaV1 and AsV341, and proposed a new genus, Alternavirus, and a new family, “*Alternaviridae”**,* to accommodate the three viruses. Subsequently, several mycoviruses were sequenced and identified as potential members of the proposed “*Alternaviridae”* family, including Fusarium poae alternavirus 1 (FpAV 1), Fusariumm graminerarum alternairus 1 (FgAV 1), Aspergillus heteromorphus alternavirus 1 (AheAV1) and Fusarium incarnatum alternavirus 1 (FiAV 1) [25,26,27,28]. Until recently, mycovirus classification into the proposed family “*Alternaviridae”* was based on phylogenetic analysis of RNA-dependent RNA polymerase (RdRp) sequences. Some genomes of viruses belonging to the proposed family “*Alternaviridae”* contained three dsRNA segments, while others comprised four dsRNA segments.

*Fusarium oxysporum* is an important plant pathogenic fungus that causes vascular wilt in a wide variety of agricultural crop species [29]. Biocontrol agents, such as plant growth- promoting rhizobacteria (PGPR) strains and nonpathogenic *F. oxysporum* strains, have proven to be effective tools for controlling plant diseases caused by *F. oxysporum* [30,31]. As more hypovirulence-associated mycoviruses have been investigated in attempts to control crop diseases, new insights and explorations regarding biological control using hypovirulence-associated mycoviruses to control plant diseases caused by *F. oxysporum* [4,32,33] have become available. To our knowledge, six mycoviruses have been reported in *F. oxysporum*. These include Fusarium oxysporum chrysovirus 1 (FoCV1), Fusarium oxysporum f.sp. dianthi mycovirus 1 (FodV1), classified as family *Chrysoviridae*; Fusarium oxysporum f. sp. dianthi mitovirus 1 (FodMV1), classified as family *Narnaviridae*; Fusarium oxysporum f. sp. dianthi hypovirus 2 (FodHV2), classified as family *Hypoviridae*; Hadaka Virus 1 (HadV1), classified as family *Polymycoviridae*; and Fusarium oxysporum ourmia-like virus 1 (FoOuLV1), classified as family *Botourmiaviridae* [34,35,36,37,38,39]. Only FodV1 and FoOuLV1 are known to cause hypovirulence [39,40].

A collection of *F. oxysporum* isolates (BH19) were obtained from fusarium wilt-diseased stems of *Lilium brownii* in the Liaoning province of China. Our preliminary study showed that a similar banding pattern of dsRNA may exist in this isolate. Sequencing and analysis of this dsRNA showed that it corresponded with a novel alternavirus—the first to be described that infects a *F. oxysporum* strain—which was provisionally named Fusarium oxysporium alternavirus 1 (FoAV1). Accordingly, we conducted genome characterization to elucidate the molecular features of FoAV1, investigate its impact on virulence and understand its biological properties in *F. oxysporum*.

## 2. Materials and Methods

### 2.1. Strains and Culture Conditions

Seven isolates of *F. oxysporium* (BH19, BH3, BH19-4V, BH19-3V, MC-1, MC-1-4V and MC-1-3V) were used in the study. Strains BH19 and BH3 were isolated from a fusarium wilt-diseased stem of *Lilium brownii* in Liaoning province of China in 2019. Strains BH19-4V and BH19-3V were single-ascospore isolates of strain BH19. MC-1 was a *F. oxysporum* f. sp. *momordicae* strain incorporating a hygromycin-resistance gene (hygromycin B phosphotransferase), which has a normal colony morphology and high virulence in its hosts. Strains MC-1-4V and MC-1-3V were derivative strains isolated from MC-1 following confrontation training with BH19-4V and BH19-3V. All strains were stored at −80 °C in glycerol and cultured on potato dextrose agar (PDA) medium at 28 °C. For dsRNA extraction, mycelium was cultured on a PDA plate covered with cellophane membranes at 28 °C for 4–5 days.

### 2.2. DsRNA Extraction and Purification

The extraction of dsRNA was performed as described previously [39]. Strains were grown for 4–5 days on cellophane membranes on PDA medium. Fresh mycelia (1–2 g) were harvested to isolate dsRNA by selective absorption using cellulose powder CF-11 (Sigma–Aldrich, St. Louis, MO, USA), with nucleic acid co-precipitator added to improve the yield of dsRNA. The dsRNA was treated with RNase-free DNase I and S1 nuclease (Takara, Dalian, China). The dsRNAs were electrophoresed on a 1% agarose gel, stained with ethidium bromide, and visualized with a gel documentation and image analysis system (InGenius LhR, Syngene, UK).

### 2.3. cDNA Synthesis, Cloning, and Sequencing

The purified dsRNA samples were used as a template for cDNA synthesis. cDNA synthesis and cloning were conducted according to the methods described previously [41]. A cDNA library was constructed using TransScript One-Step gDNA Removal and cDNA Synthesis SuperMix (^®^TransGen Biotech, Beijing, China) according to the manufacturer’s instructions. To obtain initial sequence clones, a random primer (RACE3RT) was used for RT-PCR amplification. Partial gap sequences between the initial sequences were filled by RT-PCR with sequence-specific primers designed from the obtained sequences. For the 5′ and 3′ terminal sequences cloning, an anchor primer PC3-T7 loop was ligated to the dsRNA using T4 RNA ligase and used for the RT reaction. Then, a PC2 primer (designed based on the corresponding sequence of the PC3-T7 loop) and sequence-specific primers (designed based on the proximal regions sequences) were used for the amplification of terminal sequences. All PCR products were cloned into a pMD18-T vector (Takara, Dalian, China), which was then transformed into *E. coli* DH5α cells (Takara, Dalian, China) for sequencing. To achieve high-quality consensus sequences, each nucleotide of full-length cDNA was sequenced at least three times. All the primers used for cDNA cloning and sequencing are listed in Appendix A.

### 2.4. Sequence and Phylogenetic Analysis

Open reading frames (ORFs) and conserved domains were predicted using an ORF finder and CD-search on the NCBI website (http://www.ncbi.nlm.nih.gov) and motifs scan website (http://www.genome.jp/tools/motif/). Multiple sequence alignments were performed using sequences of known alternavirus and the CLUSTALX (2.0) program [42]. Phylogenetic trees were constructed using MEGA7 software and generated by the maximum-likelihood (ML) method with 1000 bootstrap replicates [43].

### 2.5. Horizontal Transmission of Hypovirulence Traits

To assess the potential horizontal transmission of hypovirulence traits of strain BH19, dual culturing technology was used as described previously [44]. Isolate BH19 and virus-free isolate MC-1, which is hygromycin resistant, were grown separately for 4–5 days on PDA medium, and then co-cultured for a further 4–5 days on PDA fresh plates using mycelium blocks of each isolate kept in close proximity (10–15 mm) to one another. Mixed mycelium blocks were then transferred to fresh PDA plates containing hygromycin, and mycelial growth resulting from co-cultivation were termed the derivative strains. Viral transmission in these strains was evaluated based on dsRNA extraction or RT-PCR detection.

### 2.6. RNA Extraction and RT-PCR Detection

To extract total RNA, strains were grown for 4–5 days on cellophane membrane overlying PDA, and fresh mycelia were harvested and ground to a powder in liquid nitrogen. Total RNA was prepared using an RNA reagent (Newbio Industry, Wuhan, China), according to the manufacturer’s instructions.

For RT-PCR detection of virus-transmitted strains, first-strand cDNAs were synthesized using TransScript One-Step gDNA Removal and cDNA Synthesis SuperMix (^®^TransGen Biotech, Beijing, China) according to the manufacturer’s instructions. Then, PCR amplification was performed using specific oligonucleotide primers (list in Appendix A), which were designed based on the sequences of FoAV1 dsRNA. All the amplicons were identified following electrophoresis in 1.5% agarose gels.

### 2.7. Virulence Assay

The virulence of virus-transmitted strains was assessed by pot experiments on bitter gourd plants, as described previously [39]. For these experiments, bitter gourd seedlings were grown in boxes containing sterile soil, and 10 mL spores (10^7^ mL^−1^) from the various *F. oxysporum* strains were inoculated into melon roots when the plants had grown to the second or third leaf stage. Control plants were inoculated with water. All inoculated plants were monitored for the development of typical wilt symptoms, and disease symptoms were photographed. All experiments were repeated twice.

## 3. Results

### 3.1. DsRNA Segments in F. oxysporum Strain BH19

The *F. oxysporum* strains BH19 and BH3 were both isolated from a fusarium wilt-diseased stem of *L. brownii* in Liaoning Province, China. Compared to BH3, the colony morphology of BH19 was clearly defined with a small aerial mycelium (Figure 1A). Strain BH19 contained four dsRNA segments (named dsRNA1, dsRNA2, dsRNA3 and dsRNA4) while no dsRNAs were found in strain BH3 (Figure 1B). The four dsRNA bands had estimated sizes of 3.5 kbp (dsRNA1), 2.5 kbp (dsRNA2), 2.3 kbp (dsRNA3) and 1.8 kbp (dsRNA4), respectively. The segments were confirmed to be dsRNA in nature based on resistance to digestion with DNase I and S1 nuclease.

### 3.2. Molecular Characterization of FoAV1 in Strain BH19

Following cDNA cloning and sequencing, the complete sequences of the four dsRNA segments found in strain BH19 were obtained, comprising a virus nominated as Fusarium oxysporum alternavirus 1 (FoAV1). The complete sequence of FoAV1 dsRNA1 (GenBank Accession No. MT659125) was 3447 nucleotides (nt) long with a GC content of 56.6%, and it contained one ORF (ORF1) that initiated at position nt 629 and terminated at position nt 3385. Based on the universal genetic code, this sequence potentially encoded 917 amino acid (aa) residues with a calculated molecular mass (*Mr*) of 103.4 kDa. FoAV1 dsRNA2 (GenBank Accession No. MT659126) was 2677 nt in length with a GC content of 56.8%. It contained one ORF (ORF2) that initiated at position nt 66 and terminated at nt 2588, encoding 839 aa residues with an *Mr* of 92.8 kDa. FoAV1 dsRNA3 (GenBank Accession No. MT659127) was 2382 nt in length with a GC content of 53.9%. It contained one ORF (ORF3) that initiated at position nt 71 and terminated at nt 2260, encoding 728 aa residues with an *Mr* of 78.5 kDa. FoAV1 dsRNA4 (GenBank Accession No. MT659128) was 1873 nt in length with a GC content of 55.7%. It contained one ORF (ORF4) that initiated at position nt 152 and terminated at nt 1318, encoding 387 aa residues with an *Mr* of 42.6 kDa. Additionally, all the four dsRNA segments contained poly (A) tails (Figure 2).

Using the homology search on BLASTP, FoAV1 ORF1 encoded a protein which was closely related to RdRps of alternaviruses, including Aspergillus foetidus dsRNA mycovirus (67.72%), Fusarium poae alternavirus 1 (67.61%) and Alternaria alternata virus 1 (66.63%) (Table 1); FoAV1 ORF2- and ORF3-encoded proteins were also only closely related to their equivalents. Alternavirus ORF2 and ORF3 proteins showed similar identities to those predicted for ORF1 (Table 2 and Table 3). FoAV1 ORF4 encoded a protein that had no significant similarities with any other protein in the NCBI database. These results indicated that FoAV1, which was isolated from *F. oxysporum* strain BH19, contained four dsRNAs segments and was thus a new member of the proposed *“Alternaviridae”* family.

Further similarities between FoAV1 and other members of the proposed family *“Alternaviridae”* were investigated using conserved domain database searches together with the multiple protein alignment of related proteins. The obtained results suggested that FoAV1 ORF1, which putatively encode an RdRp, and equivalent proteins encoded by other members of the proposed family *“Alternaviridae”,* contained several conserved amino acid motifs (Figure 3). Similar conservation of several amino acid motifs was obtained when the proteins predicted from the sequences of FoAV1 ORF2 and ORF3 and related alternaviruses were compared (Appendix A). There were no significant similarities between the aa sequence of the protein predicted from FoAV1 ORF4 and the protein sequences predicted from other alternaviruses that contained four dsRNAs (Appendix A).

### 3.3. Phylogenetic Analysis of FoAV1

To examine the phylogenetic relationship between FoAV1 with other mycoviruses, a phylogenetic tree was constructed using a maximum likelihood method based on the RdRp (ORF1) sequences of FoAV1, proposed *“Alternaviridae”* family members and other related viruses from the families *Ourmiaviridae*, *Partitiviridae*, *Megabirnaviridae*, *Totiviridae* and *Chrysoviridae*. The results indicated that FoAV1 clustered with members of the proposed *“Alternaviridae”* family but that it was not a strain or isolate of a previously described alternavirus. The proposed *“Alternaviridae”* family is itself clustered into four clades (clades Ⅰ, Ⅱ, Ⅲ and Ⅳ) (Figure 4A). Phylogenetic trees based on all the alternavirus ORF2 and ORF3 sequences, including those from FoAV1, showed similar clustering into four clades, as illustrated in Figure 4B,C, respectively.

### 3.4. Untranslated Regions Sequences Analysis of FoAV1

The coding sequences of all four FoAV1 dsRNA segments were flanked by two untranslated regions (UTRs) at the 5′- end and 3′- termini, excluding poly (A) tails. The FoAV1 5′- and 3′-UTR sequences of all four genomic dsRNAs were strictly conserved but shared little conservation with equivalent sequences of other alternaviruses which, within the clades, shared limited conservation with each other (Figure 5A,B).

### 3.5. Elimination of FoAV1 dsRNA from F. oxysporum Strain BH19 during Subculture

During passage of the *F. oxysporum* strain BH19, it was discovered that some isolates (nominated BH19-3V), though not all (nominated BH19-4V), spontaneously lost the FoAV1 dsRNA4 genomic segment. The absence of FoAV1 dsRNA4 in BH19-3V compared to BH19-4V was confirmed following dsRNA extraction and agarose gel electrophoresis (Figure 6C), as well as by RT-PCR amplification of a fragment of FoAV1 dsRNA4 in BH19-4V (Figure 6D). The colony morphology and growth rate of *F. oxysporum* strains BH19-3V and BH19-4V were not significantly different (Figure 6A,B).

### 3.6. Effects of FoAV1 on the Virulence of F. oxysporum f. sp. momordicae

Virulence testing of *F. oxysporum* on *L. brownii* plants proved difficult, so FoAV1 was transmitted from the BH19-3V and BH19-4V strains to *F. oxysporum* f.sp. *momordicae* MC-1 to determine any effects on virulence. Using dual culture, two derivative strains, MC-1-3V and MC-1-4V, were generated, and virus infection with, respectively, three and four genomic segments, was verified (Figure 7B). Additionally, RT-PCR amplification was conducted (Figure 7C). The colony morphology of strains MC-1-3V and MC-1-4V did not significantly differ from the wild-type strain MC-1 (Figure 7A).

Fusarium wilt, caused by *F. oxysporum*, results in systemic infection and is characterized by a typical symptom: complete plant wilt. In order to determine whether the presence of FoAV1 reduced the virulence in *F. oxysporum* f. sp. *momordicae* strains, the strains MC-1, MC-1-3V and MC-1-4V were inoculated into bitter gourd seedlings at the stage of two- or three-leaf growth. Virulence was assessed before, during, and after the presentation of typical wilt symptoms (Figure 8). The results showed that bitter gourd plants inoculated with the virus-free MC-1 strain exhibited obvious dryness and wilting, while the plants inoculated with the MC-1-3V or MC-1-4V strains remained generally healthy with few or no symptoms (Figure 8).

## 4. Discussion

In this study, we identified and characterized a novel mycovirus, FoAV1, from *F. oxysporium* strain BH19, which is a causal agent of fusarium wilt disease in *Lilium brownii*. The genome of FoAV1 contains four dsRNA segments (dsRNA1, dsRNA2, dsRNA3 and dsRNA4) and a single open reading frame (ORF) in each dsRNA segment, namely: ORF1, ORF2, ORF3 and ORF4. BLAST searches revealed that ORF1, ORF2 and ORF3 were all similar in sequence to the ORFs predicted for several alternaviruses. We proposed that FoAV1 be considered a novel member of the proposed family *“Alternaviridae”*.

With the advent of low cost, next-generation sequencing, there has been a dramatic increase in the generation of complete mycovirome sequences [27,45]. Numerous mycoviruses have been described from different families, infecting many phytopathogenic fungi and entomopathogenic fungi [2,8,46]. However, among the hundreds of mycoviruses described so far, only six mycoviruses have been identified in *F. oxysporum*, an important phytopathogenic species [34,35,36,37,38,39], with no reports of any viruses belonging to the proposed family *“Alternaviridae”.* FoAV1 is the first alternavirus described to infect *F. oxysporum*.

The members of the proposed family *“Alternaviridae”* were all isolated from fungi, and clustered mainly according to homology BLAST, multiple alignment and phylogenetic analysis [24,27,28]. Until now, only eight mycoviruses have been proposed as alternaviruses of the family *“Alternaviridae”*—and these were clustered into one branch, with no classification at the genus level [21,23,24,25,26,27,28]. The RdRp sequence of FoAV1 was closely related to the known proposed alternavirus, and FoAV1 contained similar conservation of several amino acid motifs, clustered into one branch by phylogenetic analysis with other alternaviruses. Additionally, with the appearance of FoAV1, the proposed family *“Alternaviridae”* was itself clustered into four clades through phylogenetic analysis based on all alternavirus RdRps, ORF2 and ORF3 sequences and the 5′and 3′-UTR alignment. Clade Ⅰ contained Alternaria alternata virus 1 (AaV1) and Stemphylium lycopersici mycovirus (SlV); clade Ⅱ contained Aspergillus foetidus dsRNA mycovirus (AfV), Aspergillus hetero-morphus alternavirus 1 (AheAV1) and Aspergillus mycovirus 341 mycovirus (AsV341); clade Ⅲ contained only FoAV1; clade Ⅳ contained Fusarium incarnatum alter-navirus 1 (FiAV 1), Fusarium poae alternavirus 1 (FpAV 1) and Fusarium graminearum alternavirus 1 (FgAV 1). Therefore, we suggested that FoAV1 should be considered a new member of the proposed family *“Alternaviridae”* and should also be used in further studies.

In past studies, an interesting phenomenon was discovered; namely, that several segments of viruses are dispensable for replication, and these dispensable segments are considered satellite-like RNAs [47]. Satellite RNAs may have no effect at all, while some satellite RNAs are able to play important roles in biological functions [48]. Interestingly, according to the above classification of proposed alternavirus members, all members (AaV1 and SlV) in clade Ⅰ have four segments [23], two members (AfV and AsV341) in clade Ⅱ have four segments [21,24] and all members (FiAV 1, FpAV 1 and FgAV 1) in clade Ⅳ have three segments [25,26,28]. As a member of clade Ⅲ, FoAV1 has four segments, but dsRNA4 can eliminate these during subculture, with no significant difference in the effect (colony morphology, growth rate and virulence) on the host regarding the presence or absence of dsRNA4. Therefore, the dsRNA4 may be a satellite-like RNA with no known biological function, but it may also be related to the evolution of alternaviruses and used as evidence for the proposed family *“Alternaviridae”*.

Fusarium wilt caused by *F. oxysporum* is an important disease in a wide variety of agricultural crop species [29]. Biological control is an effective method for combatting this disease [30,31]. In recent years, a large number of hypovirulent mycoviruses, such as CHV1 [13] and SsHADV1 [49], have been identified and explored as potential biocontrol agents against fungal diseases [4,32,33]. However, until now, only two mycoviruses, FodV1 and FoOuLV1, isolated from *F. oxysporum*, exerted hypovirulent effects [34,35,36,37,38,39]. FoAV1 exhibited a good biological control effect for Fusarium wilt, suggesting that it may be a good resource for controlling Fusarium wilt.

In conclusion, we reported herein that FoAV1 is a new alternavirus in *F. oxysporum*, and that FoAV1 infection can cause hypovirulence in a host. Our results not only lay a foundation for the further study of mycoviruses within the proposed family *“Alternaviridae”*, but also provide a potential agent for the biocontrol of diseases caused by *F. oxysporum*.

## Figures and Tables

**Figure 1 viruses-13-02026-f001:**
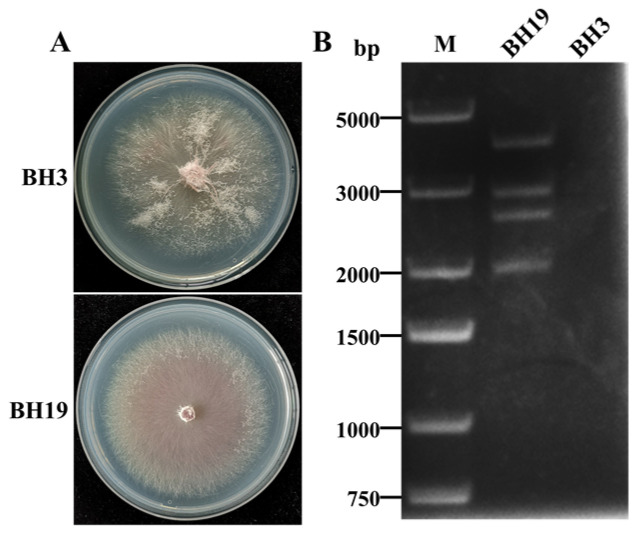
(**A**) Colony morphology of *F. oxysporum* strains BH3 and BH19. (**B**) Agarose gel electrophoresis of the dsRNA-enriched extract obtained by cellulose column chromatography from strains BH3 and BH19.

**Figure 2 viruses-13-02026-f002:**
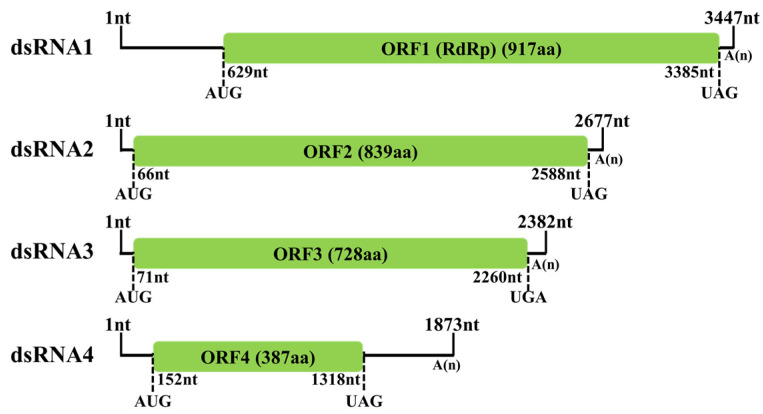
Schematic representation of the FoAV1 genome. The open reading frames (ORFs) and coding loci are indicated by a green box and single dotted lines, respectively.

**Figure 3 viruses-13-02026-f003:**
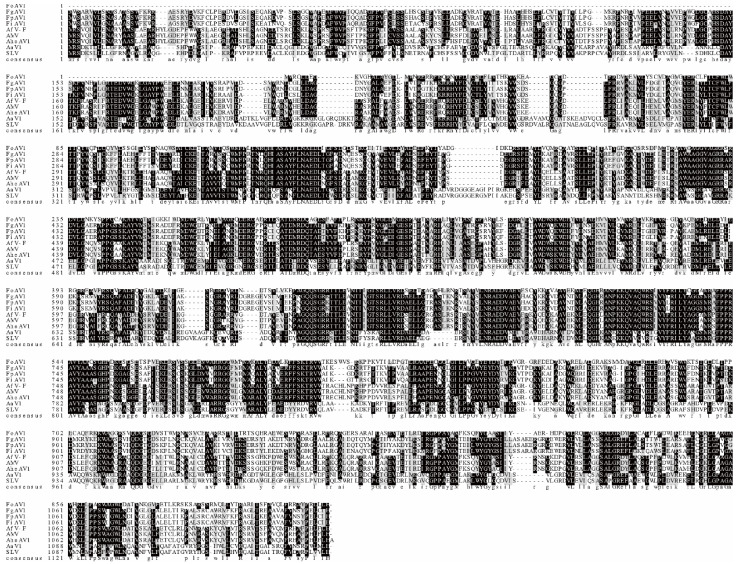
Multiple alignment of the conserved RdRp amino acid motifs encoded by FoAV1 and other *“Alternaviridae”* family members.

**Figure 4 viruses-13-02026-f004:**
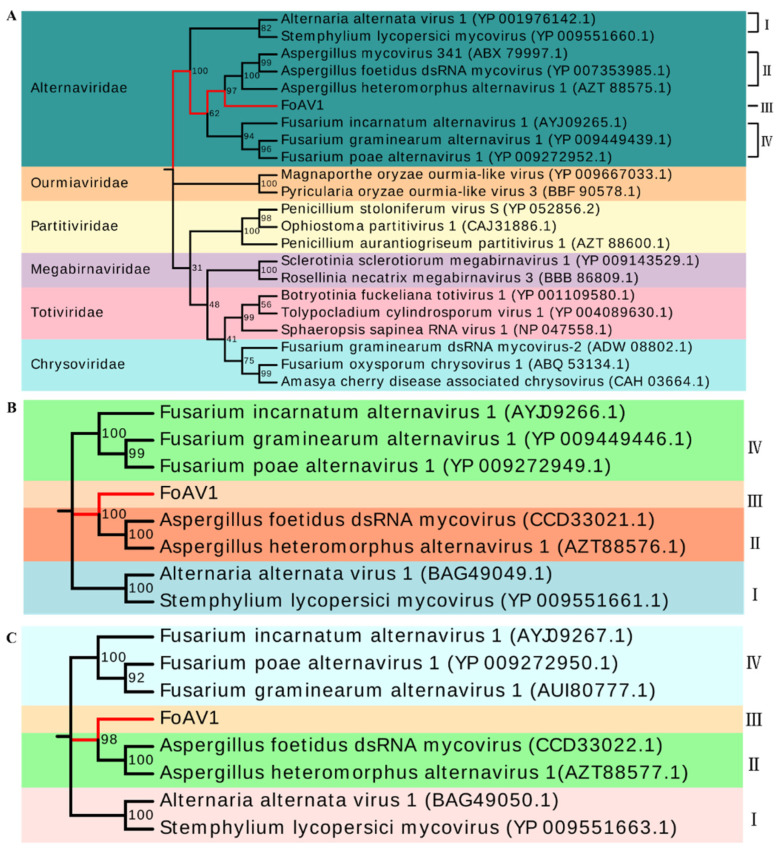
(**A**) Maximum likelihood phylogenetic trees based on the FoAV1 RdRp (ORF1) sequence, *“Alternaviridae”* family members and other related viruses. (**B**) Maximum likelihood phylogenetic tree based on the FoAV1 ORF2 sequence and other *“Alternaviridae”* family members. (**C**) Maximum likelihood phylogenetic tree based on the FoAV1 ORF3 sequence and other *“Alternaviridae”* family members. All phylogenetic analyses generated using MEGA7 with 1000 bootstrap replicates. Ⅰ to Ⅳ represent clades of the proposed *“Alternaviridae”* family. Different colors represent clustering in different families or clades.Ⅰ, Ⅱ, Ⅲ and Ⅳ represent different clades of *“Alternaviridae”* family, respectively.

**Figure 5 viruses-13-02026-f005:**
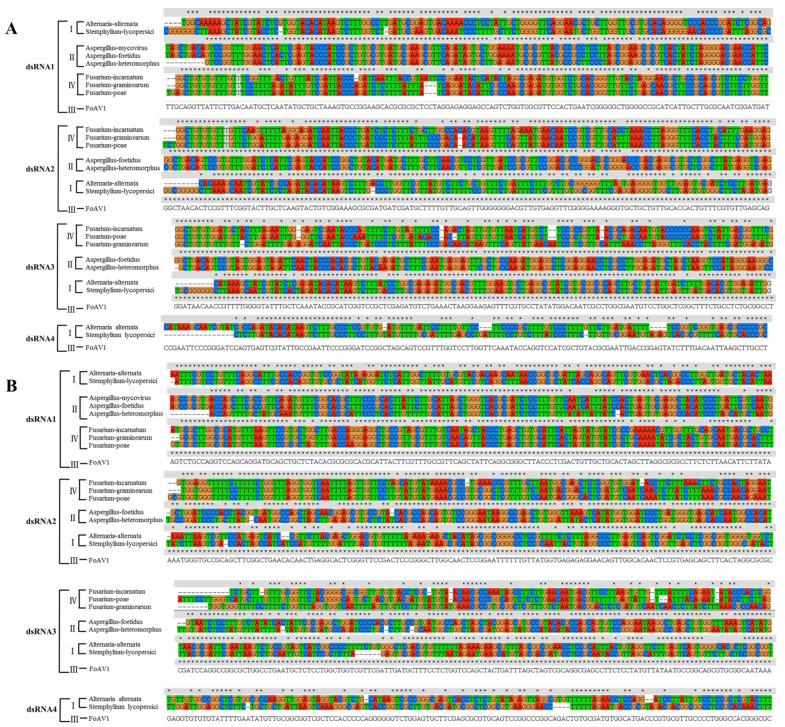
Multiple alignment of the 5′ (**A**) and 3′ (**B**)-UTRs of each FoAV1 dsRNA segment compared to equivalent alternavirus UTRs.

**Figure 6 viruses-13-02026-f006:**
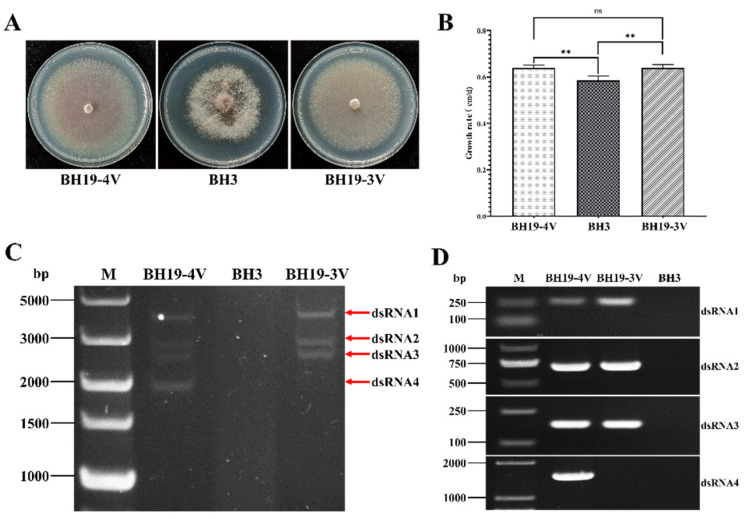
(**A**) Colony morphology of *F oxysporum* strains BH3, BH19-3V and BH19-4V. (**B**) The growth rate of strains BH3, BH19-3V and BH19-4V. “**” represents significant difference (*p* < 0.05), “ns” represents no significant difference. (**C**) Agarose gel electrophoresis of the dsRNA-enriched extract obtained by cellulose column chromatography from strains BH3, BH19-3V and BH19-4V. (**D**) The detection of each segment of FoAV1 in strains BH3, BH19-3V and BH19-4V by RT-PCR amplification.

**Figure 7 viruses-13-02026-f007:**
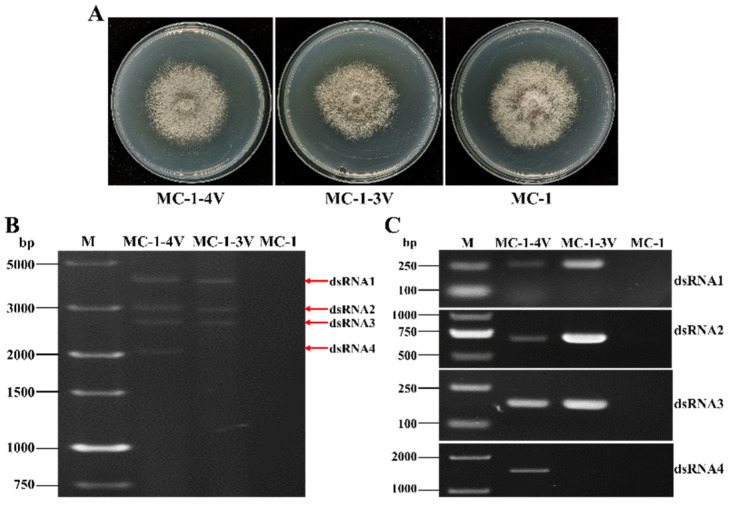
(**A**) Colony morphology of *F oxysporum* f.sp *momordicae* strains MC-1, MC-1-3V and MC-1-4V. (**B**) Agarose gel electrophoresis of the dsRNA-enriched extract obtained by cellulose column chromatography from strains MC-1, MC-1-3V and MC-1-4V. (**C**) The detection of each FoAV1 dsRNA segment in strains MC-1, MC-1-3V and MC-1-4V by RT-PCR amplification.

**Figure 8 viruses-13-02026-f008:**
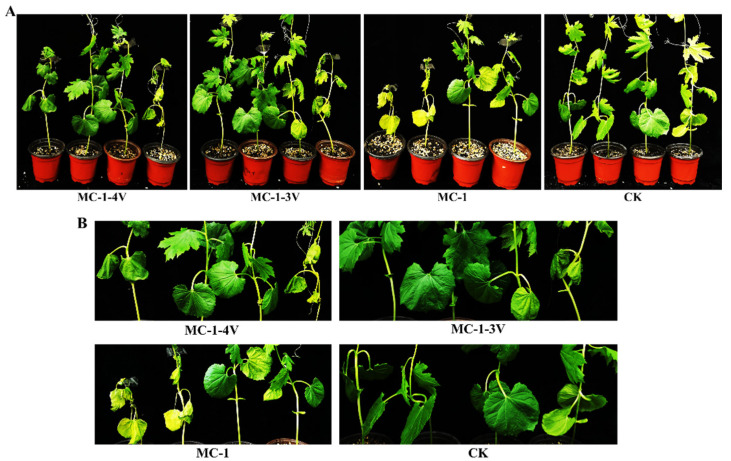
Virulence of *F. oxysporum* MC-1, MC-1-3V and MC-1-4V strains on bitter gourd plants: (**A**) Wilt symptoms on bitter gourd 15 days after inoculation with 10 mL spores (10^7^ /mL) of *F. oxysporum* MC-1, MC-1-3V and MC-1-4V strains, Control plants (CK) were inoculated with water. (**B**) Enlarged images of (Figure 8A).

**Table 1 viruses-13-02026-t001:** Identity between FoAV1 ORF1-encoded protein (RdRp) and equivalent proteins of other proposed alternaviruses.

Virus Name	Accession	Query Cover	Identity	E-Value
Aspergillus foetidus dsRNA mycovirus	YP_007353985.1	100%	67.72%	0
Aspergillus mycovirus 341	ABX79997.1	100%	67.61%	0
Aspergillus heteromorphus alternavirus 1	AZT88575.1	100%	66.63%	0
Fusarium poae alternavirus 1	YP_009272952.1	99%	48.65%	0
Fusarium graminearum alternavirus 1	YP_009449439.1	99%	48.54%	0
Fusarium incarnatum alternavirus 1	AYJ09265.1	99%	48.22%	0
Alternaria alternata virus 1	YP_001976142.1	98%	39.38%	0
Stemphylium lycopersici mycovirus	YP_009551660.1	98%	39.05%	0

**Table 2 viruses-13-02026-t002:** Identity between FoAV1 ORF2-encoded protein and equivalent proteins of other proposed alternaviruses.

Virus Name	Accession	Query Cover	Identity	E-Value
Aspergillus heteromorphus alternavirus 1	AZT88576.1	95%	40.29%	0.0
Aspergillus foetidus dsRNA mycovirus	YP_007353982.1	83%	42.28%	2 × 10^−171^
Fusarium incarnatum alternavirus 1	AYJ09266.1	62%	30.80%	1 × 10^−47^
Fusarium graminearum alternavirus 1	YP_009449446.1	66%	29.55%	7 × 10^−42^
Fusarium poae alternavirus 1	YP_009272949.1	55%	30.42%	1 × 10−^40^
Alternaria alternata virus 1	YP_001976150.1	79%	25.20%	2 × 10^−19^
Stemphylium lycopersici mycovirus	YP_009551661.1	68%	26.57%	3 × 10^−19^

**Table 3 viruses-13-02026-t003:** Identity between FoAV1 ORF3-encoded protein and equivalent proteins of other proposed alternaviruses.

Virus Name	Accession	Query Cover	Identity	E-Value
Aspergillus foetidus dsRNA mycovirus	YP_007353983.1	99%	52.26%	0.0
Aspergillus heteromorphus alternavirus 1	AZT88577.1	99%	51.78%	0.0
Fusarium graminearum alternavirus 1	AUI80777.1	97%	29.97%	7 × 10^−63^
Fusarium incarnatum alternavirus 1	AYJ09267.1	97%	30.57%	4 × 10^−62^
Fusarium poae alternavirus 1	YP_009272950.1	97%	29.97%	8 × 10^−62^
Alternaria alternata virus 1	YP_001976151.1	71%	33.33%	3 × 10^−22^
Stemphylium lycopersici mycovirus	YP_009551663.1	63%	32.36%	2 × 10^−20^

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
