# Peer review of "Molecular Characterization of the First Alternavirus Identified in *Fusarium oxysporum"

_viruses, 2021, doi:10.3390/v13102026_

Round 1

Reviewer 1 Report

 The authors characterized a novel mycovirus, FoAV1, from F. oxysporium strain BH19, which is a causal agent of fusarium wilt disease in Lilium brownii. The genome of FoAV1 contained four dsRNA segments (dsRNA1, dsRNA2, dsRNA3 and dsRNA4), and a single open reading frame (ORF) in each dsRNA segment. Homology Blast searches revealed that ORF1, ORF2, and ORF3 were all similar to the mycoviruses in the family Alternaviridae. The authors propose that FoAV1 is a novel member of the family Alternaviridae. Besides, FoAV1 is the first alternavirus described infecting F. oxysporum.

Major comments:

  1. If possible, please provide the information of the poly(A) tails of the FoAV1 dsRNA segments, since having the poly(A) tail at the 3’ ends is one of the characteristic traits as an alternavirus. Please see the paper, Aoki, et al. Virus Research 2009.
  2. If possible, please provide a couple of data for FoAV1 viral proteins, like electronic microscope photos of FoAV1 virion or viral proteins by SDS-PAGE.
  3. Please re-organize Figs. 6, and 7. The order of BH3, BH9-V, and BH-4V is not always constant. For readers, it is not convenient.

Minor points.

L12. three > four

L17 and other places. Alternaviridae (italic) > Alternavirudae (roman)

  Please correct the font for other places in the text. If it is a proposed family, it should not be italicized.

L46. Botryosphaeria. > Remove the period.

L100. F. oxysporium, fsp.?

L225. alternavirus > alternaviruses

L249. theORF3 > the ORF3 (the space needs.)

Reviewer 2 Report

The manuscript  describes a novel alternavirus in Fusarium spp., but the English and syntax of the paper is well below the standard required for publication in Viruses. I have attached an almost complete edit of the manuscript for the authors to include in a revised version of the paper.  Red deletions and yellow insertions.  I have not edited the discussion because it needs complete re-writing as it is repetitious of  a combination of the introduction and results.  I recommend that the authors seek the assistance of an English writing service before resubmission.  Annoyingly the authors wish to take it upon themselves to nominate a new family of mycoviruses proposed as the family Alternaviridae.  This is inappropriate as this is the purview of the ICTV only and not individuals.  There is some interesting novel work in the manuscript especially the elimination of the smallest genomic dsRNA segment in some isolates and a rewritten version might be acceptable for publication.  

Round 2

Reviewer 2 Report

OK to go forward for publication.